# Spatial spillover of local general higher education expenditures on sustainable regional economic growth: A spatial econometric analysis

Congying Ma[1]*, Hongchao Wu[2], Xiuhong Li[3]

**1** School of Humanities and Social Sciences, Beijing Institute of Technology, Beijing, China, **2** School of Education, South China Normal University, Guangzhou, China, **3** School of Art and Design, Guangdong University of Technology, Guangzhou, China

* 13710569783@163.com

**Data Availability Statement:** All relevant data are within the paper and its Supporting information files.

## Abstract

The ability of fair investments in local general higher education to drive sustainable regional economic growth is explored. Based on spatial theory, the exploratory spatial data analysis method is used to examine the spatial characteristics of local general higher education expenditures in China's 30 provinces from 2000 to 2021. The spatial Durbin model is employed to analyze the impact of education expenditures on regional economic growth. The results reveal that education expenditures had positive spatial autocorrelation. Education expenditures promoted regional economic growth, and the long-term effect was greater than the short-term effect. These expenditures also had a positive spillover effect, showing that strategic spatial interactions between provinces positively influence growth. The positive spillover effects nationwide and in the eastern region were significantly greater than the direct effect, whereas the spillover effects in both the middle and western regions were negative.

## Introduction

Governments and researchers worldwide have become aware of the importance of higher education in the formation of human capital, which influences sustainable economic growth, in addressing the progression and depth of environmental concerns [1]. Higher education is also closely related to the economy in various ways. It supports R&D and technological progress, which are regarded as important in most countries for enhancing growth [2]. In some countries that are shifting their economies from traditional to more sustainable, transformations in the higher education system are highly relevant to their economic development [3, 4].

As the world's second largest economy, China has experienced remarkable economic growth and urbanization since the reform and opening up in 1978; however, its sustainable development has been restricted by intensive consumption, high inputs, and high emissions, particularly in rapidly developing regions [5]. To address environmental deterioration and explore a sustainable development path, China, similar to many countries, has implemented

**Funding:** The authors received no specific funding for this work.

**Competing interests:** The authors have declared that no competing interests exist.

an innovation-driven strategy by adapting local general higher education to replace traditional input growth mechanisms [6].

In this context, local general higher education, as the main part of China's higher education system, is receiving increasing financial support from local governments. For instance, its expenditures were 1.089 trillion yuan in 2021 and accounted for 17.746% of the total education investments. Local general higher education also plays an increasingly important role in economic growth. However, local general higher expenditures (LGHEs) have significant disparities among different regions. They are significantly higher in both the eastern and western regions of the nation than in the middle region [7]. Regional differences in funding may aggravate the imbalance of economic development among regions. Hence, it is necessary to explore the following questions in this study: What kind of spatial characteristics do LGHEs present? What impact does LGHEs have on regional economic growth, and how does it affect regional growth? These issues are important for promoting educational equity and driving sustainable regional economic growth.

The top goal of this study is to present the spatial characteristics of LGHEs. In describing the key features of regional differences, many scholars have used the Gini coefficient and the Theil index to quantitatively measure regional education expenditure disparities [8, 9]. However, conventional statistical techniques have failed to distinguish the spatial differences. In this study, we investigated the spatial characteristics of LGHEs in China based on the exploratory spatial data analysis method (ESDA).

Another important purpose of this study is to explore the effect of LGHEs on regional economic growth. With the development of new economic geography, the role of education expenditure agglomeration in economic development is increasingly important [10]. However, most empirical studies that have investigated higher education expenditure (at its aggregate and disaggregate levels) and the economic growth nexus are based on the independence assumption and ignore the significant spatial interaction effects between the variables [11, 12]. This may result in estimate bias [13], which may have led to an underestimation of the impacts of higher education expenditure on economic growth in the existing research. Therefore, we further apply the spatial Durbin model (SDM) to test the impact of LGHEs on regional economic growth.

As a pioneer of spatial econometrics research, Anselin [13] put forward his spatial theory, which constitutes a comprehensive analysis of spatial econometrics. To account for the spatial spillover effects, Anselin's spatial theory was used as the theoretical basis for the spatial autocorrelation test, spatial econometric model selection and estimation. Empirically, we investigated the spatial characteristics of LGHEs in China and tested the theoretical hypothesis that LGHEs can be influenced by neighboring provinces. Then, we used the SDM to calculate the direct and spillover effects of education expenditures on growth in different regions. Furthermore, the short- and long-term effects by which LGHEs promote economic growth were explored using the DSDM.

Different from previous studies, we offer two important contributions to the literature. First, we used ESDA to investigate the global and local spatial correlation characteristics of LGHEs. The ESDA method compensates for the defect of classical statistics that ignores geospatial elements and presents differences (spatial agglomeration or heterogeneity) in spatial distribution from a geospatial perspective [14, 15]. Notably, LGHEs in a spatial agglomeration state are a premise for studying the spatial spillover of expenditures. Second, we built a more accurate SDM and dynamic spatial Durbin model (DSDM) with spatiotemporal fixed effects, which embrace spatially lagged terms, both dependent and independent variables, drawing on the complementary strengths of the spatial error model (SEM) and spatial lag panel (SLM) [13]. Since the literature concerning regional economic growth is rare, we used the models to

investigate the direct and indirect effects and short- and long-term effects of LGHEs on the economic growth of different regions in China.

The paper is structured as follows. The following section reviews the related and latest research. Section 3 introduces the selection of variables and methods of data analysis. Section 4 reports the spatial characteristics of LGHEs and analyzes the effect of LGHEs on regional economic growth. Section 5 offers conclusions and discussion. Section 6 discusses the limitations of the study and future work.

## Literature review

Research on the economic impact of education was first based on human capital theory proposed by Schultz [16] and Becker [17], which provides fundamental support for investigating how education affects economic growth. Since then, many economic growth theories have emphasized the role of education (e.g., the Solow growth model and endogenous growth theory). Many empirical studies that use education indicators as proxy variables of human capital have emerged and have found that education is positively related to economic growth [18–21]. Among them, education expenditure has been recognized as an important factor in economic growth. Typically, although neglecting to distinguish expenditure categories, a sizeable body of empirical research has focused on the effect of all education expenditure on economic growth [22, 23].

In recent decades, some studies have highlighted the effect of higher education expenditure on economic growth [24, 25]. Linear regression is a method commonly used in these studies. However, the results have been somewhat contradictory. Some have reported a positive role of education expenditures on economic growth [11, 25]. However, others that have used cross-country data [26] or state data [27, 28] have found that increased spending on higher education decreases economic growth.

In China, higher education expenditures have been found to be the driving force behind China's economic growth [29, 30]. Moreover, higher education input is an important source and driving force of technological innovation that will further promote economic growth, although there is a certain lag [31]. Furthermore, its economic growth effects are affected by gender differences in reactions to enforcement messages, as well as by governance structures and interorganizational connections [32, 33]. However, this pattern is not always consistent. For example, the scale of higher education exerted a significant positive effect on economic growth in central China, while the effect of higher education expenditure appears statistically insignificant [12].

There are numerous possible reasons for these inconsistent results. First, they may be attributed to differences in statistical methods and selected samples or the presence of endogeneity (e.g., reverse causality) and omitted variables [34, 35]. Second, an overexpanded college system may bring negative effects (e.g., unemployment, precarious work, etc.) [36]. Finally, the role of investment in higher education may be significant in other nonmonetary areas. These effects are sufficient to offset the ineffective economic growth effects. For instance, higher education institutions disseminate knowledge and increase critical thinking, which are considered necessary for establishing a sustainable society [37].

With the development of new economic geography, researchers have taken the spatial interaction effect into consideration. Some empirical evidence has shown that the level of education expenditure can be affected by the expenditures of neighboring jurisdictions [38–41]. Additionally, education spending has a direct effect on a province's own economic activity and creates interjurisdictional spillovers that influence neighboring provinces' growth [20, 42, 43].

There are two possible reasons for education funding having a spillover effect. First, notably, in economically integrated economies, most economic policies are spatially dependent. The policy choice of one country or region depends partly on the policy choices of neighboring countries or regions. Second, local expenditure in neighboring political units can be spatially interdependent due to spillovers, competition effects, or mimicking [44].

In summary, the above analysis shows that there is still uncertainty concerning the investment of LGHEs and ultimately the promotion of sustainable economic growth in China. Although the spatial spillover effects of education expenditure have gradually gained attention, the corresponding research is limited in depth and fails to explore the spatial characteristics of LGHEs and their impact on regional economic growth. Hence, based on Anselin's spatial theory, this study uses ESDA to examine the spatial characteristics of LGHEs. The SDM and DSDM are employed to analyze the impact of LGHEs on regional economic growth.

## Data and methods

### Variables

**Dependent variable.**   We selected the province-level real per capita GDP (PGDP) as the dependent variable. It has been used in many empirical examinations as a standard proxy for economic growth [45]. Additionally, it is considered a standard growth proxy for provincial examinations in China.

**Independent variable.**   We selected LGHE as the independent variable and per student expenditure as its proxy variable. Per-student expenditure is an important part of LGHEs. It reflects the educational resources that each student can receive, which is an important indicator of the fairness of education expenditure. Following human capital theory [16, 17], LGHE promotes economic growth through direct or indirect channels. Furthermore, technological innovation is an important mediating variable in indirect channels, which can improve total factor productivity and upgrade the industrial structure, ultimately promoting economic growth [31].

**Control variables.**   Previous research has covered a broad range of factors related to economic growth among Chinese provinces [46–48]. This study adopted the following four control variables: capital investment (K), industrial structure (Ind), urbanization level (Urb) and foreign direct investment (Fdi) (Table 1).

**Sample and data sources.**   We selected panel data from 2000 to 2021 in China's 30 provinces. All research data were sourced from the China Statistics Yearbook (2001–2022) and the China Educational Finance Statistics Yearbook (2001–2022). All provinces were categorized into three regions based on their location and economic situation: eastern, middle, and western. In 2022, the GDP of the eastern region reached 62.202 trillion yuan, successfully constituting half of that of the whole country. Compared to the eastern region, the middle region is developing more slowly, and the western region is the poorest [49]. All data are logarithmic and estimated at the 2000 price level. Table 2 displays the descriptive statistics for the complete dataset.

### Method

**ESDA.**   It is generally believed that education expenditure has a spatial spillover effect and spatial correlation [39, 40]. Therefore, we used the ESDA method, Stata 17 and ArcGIS 10.8 software for data analysis and adopted global Moran's I and local Moran's I to detect the spatial characteristics of LGHEs.

To test whether LGHEs in China have spatial agglomeration characteristics, the most commonly used global Moran's I is adopted for global spatial autocorrelation analysis [13]. It can

**Table 1. Control variable selection and symbol.**

| Variable | Definition | Symbol | Unit |
|---|---|---|---|
| Capital investment | proportion of fixed asset investments in provincial GDP | K | % |
| Industrial structure | proportion of the secondary industry's value-added in provincial GDP | Ind | % |
| Urbanization level | proportion of the urban population in the total population of the province | Urb | % |
| Foreign direct investment | proportion of the foreign direct investment in provincial GDP | Fdi | % |

1. Capital investment has a positive effect on economic growth. It stimulates economic growth through the release of the demand effect and forms production capacity through the supply effect to promote an increase in the output level.

2. Industrial structure, when upgraded and rationalized, can improve labor productivity, transform the mode of economic development, and promote economic growth. Economic growth will also adversely affect the state of industrial structure.

3. Urbanization promotes sustainable economic growth mainly by increasing physical capital to stimulate investments, accumulating human capital, promoting the transfer of land elements, promoting technological innovation, and improving the level of residents' consumption.

4. Foreign direct investment has the dual role of boosting production efficiency and shifting the production frontier. It promotes technological progress and catching up with the production frontier by enhancing the accumulation of physical capital and human capital; furthermore, it is a powerful driving force for economic growth.

be expressed as:

$$I = \frac{n \sum_{i=1}^{n} \sum_{j=1}^{n} W_{ij}(X_i - \overline{X})(X_j - \overline{X})}{\sum_{i=1}^{n} \sum_{j=1}^{n} W_{ij} \sum_{i=1}^{n}(X_i - \overline{X})^2} \tag{1}$$

where n is the number of provinces; X is the variable of interest, LGHE; and $W_{ij}$ is the geographic distance spatial weight matrix of provinces i and j. It is assumed that the closer province i is to province j, the greater $W_{ij}$ will be. Moran's I takes the value [–1,1]. The closer Moran's I value is to -1 or 1, the stronger the spatial negative (positive) correlation is.

Local Moran's I is employed to explore cluster patterns and spatial patterns [13]. It can be expressed as:

$$I_p = \frac{(n-1) \sum_{q=1,q \neq p}^{n} W_{pq}(X_p - \overline{X})(X_q - \overline{X})}{\sum_{q=1,q \neq p}^{n}(X_q - \overline{X})^2} \tag{2}$$

When Moran's I>0, it represents high-high or low-low cluster patterns. When Moran's I<0, it

**Table 2. Descriptive statistics.**

| Variable | Observations | M | SD | Min | Max |
|---|---|---|---|---|---|
| LnPGDP | 660 | 9.168 | 0.528 | 7.887 | 10.781 |
| LnLGHE | 660 | 9.590 | 0.389 | 8.437 | 10.953 |
| LnK | 660 | 5.538 | 1.020 | 2.636 | 8.064 |
| LnInd | 660 | 3.724 | 0.222 | 2.771 | 4.126 |
| LnUrb | 660 | 3.895 | 0.327 | 2.631 | 4.495 |
| LnFdi | 660 | 0.409 | 1.121 | -5.085 | 2.684 |

represents high-low (high value surrounded by low values) or low-high (low value surrounded by high values) cluster patterns.

**SDM.** Based on the study of Mankiw, Romer, and Weil [50], this study draws on Shioji's [51] public investment economic growth equation to construct a model of the impact of local general higher education investment on economic growth, as shown in Eq (3):

$$LnY_{it} = \beta LnX_{it} + u_i + \lambda_t + \varepsilon_{it} \tag{3}$$

Based on Eq (3), the SDM is defined as:

$$LnY_{it} = \rho \sum_{j=1}^{n} W_{ij} LnY_{it} + \beta LnX_{it} + \sum_{j=1}^{n} \theta W_{ij} LnX_{it} + u_i + v_t + \varepsilon_{it} \tag{4}$$

where $Y_{it}$ is the dependent variable; $X_{it}$ is the independent variable, LGHE, and a set of controlled variables; $\rho$ and $\theta$ are spatial lagging coefficients; $\beta$ is the parameter to be estimated; $\mu_i$ is the individual effect; $v_t$ is the time effect; and $\varepsilon_{it}$ is the random error term.

If there are spatial lag terms, the regression coefficient of the econometric model cannot directly be used to reflect the marginal effect of independent variables. To judge the spatial spillover effect of LGHEs on regional economic growth more effectively, this study applies the partial differential method to measure the direct and spillover effects in the SDM [10, 52], accounting for the spatial correlation information of the dependent variable and the independent variable. Eqs 5–7 show the direct, spillover, and total effects of the SDM.

$$Direct\ effects = \left[ (I_N - \rho W)^{-1} (\beta_k I_N + \theta_k W) \right]^{\overline{d}} \tag{5}$$

$$Spillover\ effects = \left[ (I_N - \rho W)^{-1} (\beta_k I_N + \theta_k W) \right]^{\overline{rsum}} \tag{6}$$

$$Total\ effects = \left[ (I_N - \rho W)^{-1} (\beta_k I_N + \theta_k W) \right]^{\overline{d}} + \left[ (I_N - \rho W)^{-1} (\beta_k I_N + \theta_k W) \right]^{\overline{rsum}} \tag{7}$$

where $I_N$ is an identity matrix, k represents k-th explanatory variable, and $\overline{d}$ and $\overline{r}$ sum denote two operators that can be used to calculate both the mean diagonal element and the mean row sum of the nondiagonal elements of a matrix [52].

**DSDM.** Considering that the dependent variable is affected by the previous period, we construct a DSDM that incorporates the spatial lag item of the dependent variable. This model can alleviate the endogeneity problem of the model to a certain extent [53]. It can also be used to test the existence of endogenous and exogenous interaction effects in the short term along with the long term [10]. Eq (8) presents this model.

$$LnY_{it} = \alpha \sum_{j=1}^{n} W_{ij} LnY_{i,t-1} + \rho \sum_{j=1}^{n} W_{ij} LnY_{it} + \beta LnX_{it} + + \sum_{j=1}^{n} \theta W_{ij} LnX_{it} + u_i + v_t + \varepsilon_{it} \tag{8}$$

Eqs 9–14 show the direct, spillover, and total effects of the DSDM in the short-term and

long-term scales:

$$\text{Short-term direct effects} = \left[(I_N - \rho W)^{-1}(\beta_k I_N + \theta_k W)\right]^{\overline{d}} \tag{9}$$

$$\text{Short-term spillover effects} = \left[(I_N - \rho W)^{-1}(\beta_k I_N + \theta_k W)\right]^{\overline{rsum}} \tag{10}$$

$$\text{Short-term total effects}$$
$$= \left[(I_N - \rho W)^{-1}(\beta_k I_N + \theta_k W)\right]^{\overline{d}} + \left[(I_N - \rho W)^{-1}(\beta_k I_N + \theta_k W)\right]^{\overline{rsum}} \tag{11}$$

$$\text{Long-term direct effects} = \left[(I_N - \rho W - \alpha W)^{-1}(\beta_k I_N + \theta_k W)\right]^{\overline{d}} \tag{12}$$

$$\text{Long-term spillover effects} = \left[(I_N - \rho W - \alpha W)^{-1}(\beta_k I_N + \theta_k W)\right]^{\overline{rsum}} \tag{13}$$

$$\text{Long-term total effects}$$
$$= \left[(I_N - \rho W - \alpha W)^{-1}(\beta_k I_N + \theta_k W)\right]^{\overline{d}} + \left[(I_N - \rho W - \alpha W)^{-1}(\beta_k I_N + \theta_k W)\right]^{\overline{rsum}} \tag{14}$$

## Results

### Spatial characteristics of LGHEs

**Global spatial autocorrelation.** Table 3 presents the global Moran's I values of LGHEs from 2000 to 2021. Except for 2014 to 2016 and 2019 to 2021, all other global Moran's I values are positive at above the 10% significance level. This result indicates that LGHEs are not independently distributed and have positive spatial correlation and spatial agglomeration characteristics. In other words, LGHEs are geographic neighbors; the provinces with high LGHEs are geographic neighbors, as are those with low HPEs.

**Table 3. Global Moran's I index of LGHEs.**

| Year | Moran's I | Z value | P value | Year | Moran's I | Z value | P value |
|------|-----------|---------|---------|------|-----------|---------|---------|
| 2000 | 0.194 | 2.442 | 0.015** | 2011 | 0.140 | 2.064 | 0.039** |
| 2001 | 0.191 | 2.410 | 0.016** | 2012 | 0.209 | 3.089 | 0.002*** |
| 2002 | 0.125 | 1.824 | 0.068* | 2013 | 0.223 | 3.215 | 0.001*** |
| 2003 | 0.281 | 3.418 | 0.001*** | 2014 | 0.085 | 1.549 | 0.121 |
| 2004 | 0.198 | 2.612 | 0.009*** | 2015 | 0.070 | 1.311 | 0.190 |
| 2005 | 0.209 | 2.693 | 0.007*** | 2016 | 0.085 | 1.507 | 0.132 |
| 2006 | 0.169 | 2.282 | 0.022** | 2017 | 0.111 | 1.862 | 0.063* |
| 2007 | 0.135 | 1.895 | 0.058* | 2018 | 0.113 | 1.783 | 0.075* |
| 2008 | 0.136 | 1.931 | 0.053* | 2019 | 0.063 | 1.172 | 0.241 |
| 2009 | 0.120 | 1.892 | 0.059* | 2020 | 0.085 | 1.358 | 0.175 |
| 2010 | 0.218 | 3.157 | 0.002*** | 2021 | 0.021 | 0.678 | 0.498 |

*Note*:

*$p <$10%,

**$p <$5%,

***$p <$1%.

From the overall trend, the global Moran's I values show a gradual downward trend. The highest point occurs at 0.281 in 2003, indicating that the LGHE in 2003 has the strongest spatial agglomeration. After 2003, the Moran's I values of LGHEs demonstrate a wavelike drop, indicating that since 2003, the spatial agglomeration of LGHEs has weakened.

**Local spatial autocorrelation.** Local Moran's I was employed to further reflect the evolution of cluster patterns regarding LGHEs. With the help of ArcGIS 10.8 software, we selected the four years of 2000, 2007, 2014, and 2021 as representative time nodes to draw the LISA cluster diagram of LGHEs.

Fig 1 shows that LGHEs in China are mainly low-low and high-high clustering types. Except for Gansu, which is the low-high clustering type, and Inner Mongolia, which gradually changes from the low-high to high-high clustering type, the other provinces in western China are the low-low clustering type. In addition, Beijing, Tianjin, Shanghai, Zhejiang, and Fujian are the high-high clustering type, while Guangdong, the largest economic province in China, is the high-low clustering type. This implies that the LGHPEs in most western provinces and some eastern provinces have positive spillover effects. The effects may extend from the eastern and western provinces to the middle provinces and particularly to adjacent provinces.

The above analyses show that LGHEs have spatial agglomeration and spatial heterogeneity characteristics. The provinces and numbers included in each type of agglomeration area have changed in different years and have not formed a stable spatial pattern. This can be explained

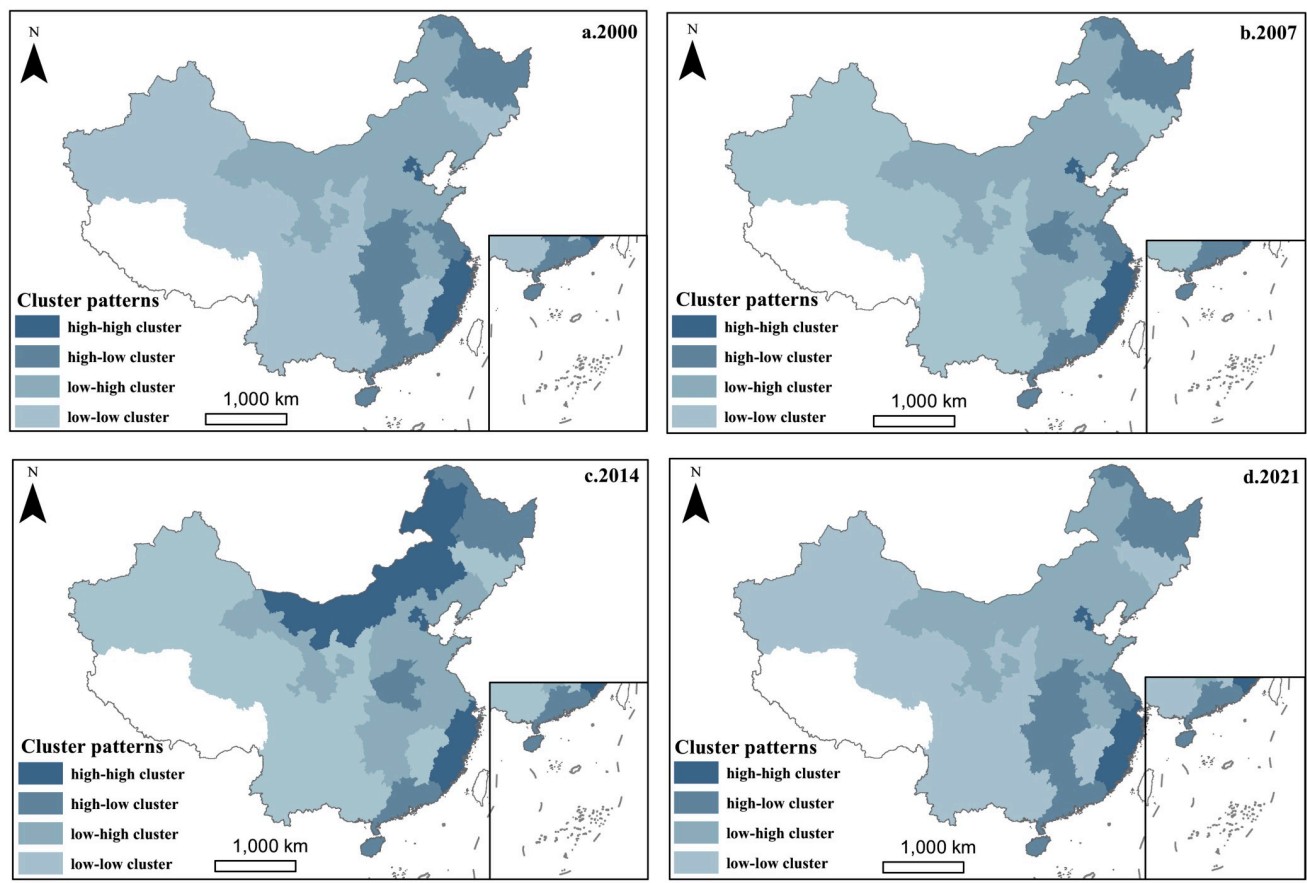

**Fig 1. LISA clustering results of LGHEs.** a. 2000, b. 2007, c. 2014, d. 2021. The map is obtained from Natural Earth (http://www.naturalearthdata.com/).

by the reality that financial sources for LGHEs in China are diversified (e.g., government financial, tuition fees and social funding) [54]. To some extent, the amount of funding may also be affected by the local economic situation and education policies [55], which have increased the instability of the spatial distribution of LGHEs.

## The impact of LGHEs on regional economic growth

**Model specification test.** As stated earlier, LGHEs have spatial autocorrelation characteristics. Therefore, a panel-based spatial measurement model is suitable for this empirical analysis. The procedure proposed by Anselin [13] and Elhorst [56] was applied here to test the appropriateness of the spatial model specification, as shown in Table 4. The Lagrange multiplier (LM) tests indicate that SEM or the SLM model with spatial effects can be used for spatial measurements. The suitability of SDM, which is the general form of SEM and SLM, needs further testing [10]. Furthermore, the results of the Wald and likelihood ratio (LR) tests favor the SDM. According to the results of the Hausman tests and China's economic characteristics, we explained the results in this study via the SDM with spatiotemporal fixed effects.

**Baseline model regression analysis.** Based on the spatial theory of Anselin [13], the SDM with spatiotemporal fixed effects is constructed for the whole country and the eastern, middle, and western areas. The regression results are presented in Table 5.

ρ is statistically significant at the 10% level, indicating that the PGDP of the whole country and the middle region has significant spatial correlations and spillover effects. A province's increase in PGDP by one unit can be expected to enhance the PGDP in neighboring provinces because of a positive spillover effect. This finding suggests that spatial interaction is a crucial factor in driving economic growth through the regional integration strategy.

The coefficients of LGPE are significantly positively correlated nationwide and in the eastern region, indicating that investment in local general higher education promotes economic growth. The results showed that a 1% increase in LGPE is estimated to increase economic output by 0.128% and 0.058% nationwide and in the east, respectively, further confirming that human capital investment is positive and significant for economic growth. However, the coefficients of LGPE are insignificantly positively or negatively correlated for the middle and western regions. A possible explanation is that the use of LGHE is inefficient, and there is a lack of

**Table 4. Test results of the spatial panel data model.**

| Statistical quantities | Nationwide | Eastern | Middle | Western |
|---|---|---|---|---|
| LM-lag | 57.338*** | 64.134*** | 6.541** | 4.352** |
| LM-error | 32.502*** | 0.775 | 0.116 | 0.214 |
| LM-lag (robust) | 53.457*** | 63.628*** | 6.482** | 4.288** |
| LM-error (robust) | 28.621*** | 0.269 | 0.058 | 0.151 |
| Wald-lag | 4.39** | 14.14*** | 3.81* | 28.46*** |
| Wald-error | 8.01*** | 15.27*** | 3.42* | 29.41*** |
| LR-lag | 160.06*** | 110.59*** | 48.02*** | 55.00*** |
| LR-error | 157.09*** | 110.25*** | 62.69*** | 54.69*** |
| Hausman test | 22.13*** | 187.28*** | 200.79*** | 4.92 |

*Note*:

*$p < 10\%$,

**$p < 5\%$,

***$p < 1\%$.

**Table 5. Estimation results of the SDM.**

| Variable | Nationwide | Eastern | Middle | Western |
|---|---|---|---|---|
| LnLGHE | 0.128*** | 0.058** | 0.038 | -0.036 |
| LnK | -0.082*** | -0.173*** | -0.188*** | 0.114*** |
| LnInd | 0.167*** | 0.197*** | -0.013 | -0.460*** |
| LnUrb | -0.084*** | -0.103*** | 0.272** | -0.067* |
| LnFdi | 0.024*** | 0.015 | 0.017 | 0.001 |
| W×LnLGHE | 0.127*** | 0.188*** | -0.179* | -0.447*** |
| W×LnK | 0.173*** | 0.457*** | -0.153** | 0.250*** |
| W×LnInd | 0.596*** | 0.182 | -0.361*** | 0.107 |
| W×LnUrb | 0.181*** | -0.093** | 0.547*** | -0.510** |
| W×LnFdi | -0.052*** | -0.072*** | 0.083*** | -0.004 |
| Spatial Rho (ρ) | 0.338*** | 0.099 | 0.214*** | -0.109 |
| $R^2$ | 0.136 | 0.006 | 0.192 | 0.398 |
| Log-likelihood | 807.264 | 696.898 | -489.333 | 333.315 |

*Note*:

*$p < 10\%$,

**$p < 5\%$,

***$p < 1\%$.

high-quality teachers. This results in the human capital effect of local general higher education not being fully utilized.

While not our primary focus, it is worth mentioning the results related to certain controlled variables. The coefficients of some controlled variables also show significant positive correlations, indicating that human capital is not the only economic growth factor; other factors may also promote economic growth. Moreover, some control variables also have positive spatial spillover effects. In particular, the industrial structure in the whole country has significant positive spatial spillover effects on economic growth, and the spatial spillover effects are strong.

**Estimation of the spatial spillover effects.**   Due to the large differences in resource endowments in various regions of China, the spillover effects of LGHEs on regional economic growth are also obviously heterogeneous. We further used the SDM to measure spillover effects nationwide and in the eastern, central, and western areas, as presented in Table 6.

At the national level, LGHE has a significant positive impact on PGDP through direct, spillover, and total effects. Among them, the positive spillover effect of education expenditures is approximately 1.79 times the direct effect, indicating that the spatial spillover effect of education expenditures in neighboring provinces has an important effect on local economic growth. Regarding the three regions, the spatial effect presents regional differences. The spillover effects are significantly positive in the eastern region. If the eastern region's spending increases by 1%, local economic growth may directly increase by 0.065%. Additionally, neighboring provinces may experience a 0.212% increase. Considering the spillover effect, a 1% increase in the LGHE could result in a 0.277% increase in the eastern region, which is a reasonable estimation.

However, the spillovers are significantly negative for the middle and western regions, indicating that a province's investments in education expenditures have a negative effect on neighboring provinces' economic growth. The negative spillover effect is as follows: western > middle. Possible explanations include a substantial loss of human resources in the middle and western regions. Investments in local general higher education in local provinces

**Table 6. Spatial effect decomposition of SDM.**

| Effect | Nationwide | Eastern | Middle | Western |
|---|---|---|---|---|
| Direct effect | 0.140*** | 0.065** | 0.024 | -0.024 |
| Spillover effect | 0.250*** | 0.212*** | -0.198* | -0.404*** |
| Total effect | 0.390*** | 0.277*** | -0.174 | -0.429*** |

*Note*:

*p <10%,

**p <5%,

***p <1%.

will improve the local environment to a certain extent and attract talent inflow from neighboring provinces, thereby reducing neighboring provinces' economic growth.

**Estimating the short- and long-term effects.** The DSDM was further used for estimation, as shown in Table 7.

Generally, the results show that investment in LGHEs in the short and long term can promote economic growth nationwide and in the eastern region, and the long-term effect is greater than the short-term effect. Among them, the short-term spillover effect nationwide is not significant, while the long-term effect is positive and significant. In the long run, the negative effect of the western region decreases. Therefore, we should note that the impact of education spending on economic growth is not immediate but requires the time for a generation to complete high school, plus time to find and learn a new job. Moreover, the effects of education are cumulative [57]. Therefore, the government must conduct short- and long-term planning when investing in LGPE to maximize the effectiveness of education expenditure in promoting sustainable regional economic growth.

**Robustness test.** The method of adding control variables to the original model is used to test the robustness of the aforementioned SDM with spatiotemporal fixed effects. According to previous empirical literature on economic growth [58, 59], railways improve regional accessibility, promote the rapid flow of production factors, enhance the effectiveness of spatial resource distribution, and realize the effect of factor agglomeration, which is an important factor affecting economic growth. We selected the density of the railway network (Way) as the added control variable, and the proportion of mileage of railway network operation in each

**Table 7. Short- and long-term effects decomposition of DSDM.**

| | Effect | Nationwide | Eastern | Middle | Western |
|---|---|---|---|---|---|
| Short-term | Direct effect | 0.107*** | 0.062** | 0.010 | -0.037 |
| | Spillover effect | 0.117 | 0.203*** | -0.140 | -0.478*** |
| | Total effect | 0.223** | 0.266*** | -0.129 | -0.514*** |
| Long-term | Direct effect | 0.124*** | 0.069** | 0.008 | -0.010 |
| | Spillover effect | 0.333* | 0.228*** | -0.143 | -0.383*** |
| | Total effect | 0.457** | 0.297*** | -0.135 | -0.393*** |

*Note*:

*p <10%,

**p <5%,

***p <1%.

**Table 8. Robustness test results of SDM.**

| Variable | Nationwide | Eastern | Middle | Western |
|---|---|---|---|---|
| LnLGHE | 0.127*** | 0.048* | -0.023 | -0.029 |
| LnK | -0.082*** | -0.176*** | -0.177*** | 0.115*** |
| LnInd | 0.153*** | 0.162** | -0.075 | -0.466*** |
| LnUrb | -0.097*** | -0.104*** | 0.134 | -0.091** |
| LnFdi | 0.023*** | 0.015 | 0.016 | 0.001 |
| LnWay | -0.018 | 0.047* | 0.129* | -0.004 |
| W×LnLGHE | 0.123** | 0.213*** | -0.274*** | -0.482*** |
| W×LnK | 0.156*** | 0.458*** | -0.070 | 0.219*** |
| W×LnInd | 0.529*** | 0.220 | -0.428*** | 0.308 |
| W×LnUrb | 0.148*** | -0.116*** | 0.212 | -0.344* |
| W×LnFdi | -0.053** | -0.073*** | 0.087*** | -0.009 |
| W×LnWay | 0.157*** | -0.037 | 0.609*** | 0.257** |
| Spatial Rho (ρ) | 0.324*** | 0.076 | 0.138 | -0.098 |
| $R^2$ | 0.290 | 0.001 | 0.153 | 0.394 |
| Log-likelihood | 811.599 | 696.898 | -489.3327 | 336.272 |

*Note*:

*$p <10\%$,

**$p <5\%$,

***$p <1\%$.

province in relation to provincial area is used as a proxy variable. The results are illustrated in Table 8.

A comparison of Tables 5 and 8 shows that after adding the density of the railway network, the direction and significance of coefficients seldom change in the explanatory variables, interaction terms of explanatory variables and spatial weight matrix, the control variables, and interaction terms of control variables and the spatial weight matrix. These results confirm that the empirical analysis results of the effect of the LGHE on regional economic growth are robust.

## Conclusions and discussion

Based on the spatial theory of Anselin [13], we used ESDA to verify the theoretical hypothesis that LGHEs have spatial autocorrelation and further estimated the SDM and DSDM with spatiotemporal fixed effects to study the impact of LGHEs on regional economic growth.

The empirical results show that LGHEs have positive spatial autocorrelation. Specifically, most western provinces, Beijing–Tianjin, and the coastal provinces in southeast China have positive spillover effects. The findings show that the spillover effects of interprovincial interactions are positive, supporting prior studies [60]. Both theory and causal observations suggest that expenditure spillovers are a widespread feature of many services provided by local governments [54]. We assume that the spatial spillover of LGHEs occurs through the following mechanisms: (a) the transferability and portability of knowledge itself; (b) the cross-regional flow of the population; and (c) commodity trade and investment exchanges [29].

The estimated results also suggest a noteworthy and affirmative correlation between LGHEs and economic progress, as the long-term effects were greater than the short-term ones, which reinforces previous research indicating that augmenting investments in local general higher education positively influences economic growth [18, 61]. However, our estimates are

larger than those of previous studies [60, 62]. One explanation is that the results may be underestimated when no spatial effect is considered. Although the literature on the impact of higher education spending on economic growth contains mixed findings, there is strong evidence that such spending can have long-lasting and robust effects on promoting growth. The new evidence in this study also suggests that investing in local general higher education has positive spillover effects.

The new evidence in this study also suggests that investing in local general higher education has positive spillover effects. Furthermore, the estimated results indicate that spillover effects are positive and significant for economic growth nationwide and in the eastern region. Interestingly, the spillover effect in both the central and western regions is negative. The above analyses highlight the important role of the spatial spillover effects of LGHEs on regional economic growth, which further verifies the clustering effect and competitive effect of the funds [29].

Based on the findings above, we draw the following policy implications. The suggestions are applicable not only for China but also for other countries.

First, local governments need to invest in local general higher education to improve research capabilities and promote technological innovation so that local general higher education can make significant contributions to economic development. It is also necessary to introduce preferential policies, including taxation policies, encourage social funding in higher education, and advance the diversification of the funding sources of expenditures. What is clear is that local general higher education needs stable and reliable public financial input. Diversified sources of funds should not be a substitute for government input but an important and appropriate—although limited—supplement. Furthermore, although the cost-sharing mechanism alleviates the government's funding pressure, it may be detrimental to participation in the expansion of higher education, which is a dilemma worth exploring in further studies.

Second, due to the obvious regional inequalities in LGHEs, China's central government must improve the financial subsidy mechanism to reduce regional differences and promote regional equity in LGHEs. Policies that promote both quality and equity in education are crucial for students and the sustainable progress of society [63, 64]. In fact, ensuring a more equitable distribution of education not only benefits educational development but also aligns with the objective of economic growth [65]. Thus, we must increase investments, eliminate some institutional barriers and establish a lasting investment mechanism for LGHEs in regions with lower expenditures to promote sustainable regional economic growth [66].

Finally, education policies should consider the spatial effects of neighboring regions. Local governments can guide regions to conduct an orderly, adequate exchange of talent, funds and technology, establish a regional integration mechanism, broaden channels for the spillover of education expenditures, and strengthen the cooperation between universities to achieve resource sharing and exchange and increase spatial spillover effects for education expenditures. Additionally, interregional educational integration policy can promote the balanced development of human capital among regions through spillover effects. Furthermore, regions with positive spillover effects on growth should not only increase investments in education and promote economic development in their local regions but also establish higher education resource clusters to accelerate the spillover of education expenditures and drive the development of neighboring areas. Specifically, the spatial spillover effect of LGHEs on economic growth should be brought into play to drive regions where the total effects are negative (e.g., the middle and western regions in China), ultimately narrowing regional economic differences and promoting coordinated and sustainable regional economic growth.

The contributions of this paper are as follows. First, we use EDSA to analyze the spatial characteristics of LGHE and verify the theoretical hypothesis that LGHEs have spatial autocorrelation. Second, spatial econometric analysis compensates for the ignorance of the spatial spillover effects of variables when exploring the relationship between the LGHE and regional economic growth. Finally, this paper expanded the application of spatial theory in the field of economic growth.

The findings are of great value to informing policymakers to make educational policies and regional development strategies more effectively consider the short- and long-term economic impacts of LGHEs, regional inequalities in LGHEs and the spatial effects of neighboring regions, which can further promote fair investments in local general higher education, reduce regional disparities, and drive sustainable regional economic growth. They may also have significant implications worldwide for optimizing the allocation of scarce resources for education expenditures and the promotion of sustainable regional economic growth. In particular, our findings may provide useful suggestions for countries with regional economic growth disparities or that are experiencing the transformation of economic development patterns (e.g., from traditional to sustainable growth).

## Limitations and future work

There are still some limitations in this study. We focus on LGHEs as the independent variable and do not consider the influence of other educational factors (e.g., per student schooling years) on economic growth. In fact, human capital and physical capital jointly affect economic growth [18]. Therefore, educational factors should be considered comprehensively to improve the scientific rationality of measuring the impact of local general higher education on economic growth. This study also lacks comparative studies across multiple countries.

In the future, there is still much pertinent work to do. For instance, the influence of changes in higher education expenditure inequality on economic growth should be discussed across multiple countries. Future work may also explore how spillover effects of other educational factors (e.g., per student schooling years) affect the sustainable growth of regional economies. Researchers can further expand the analysis of regional spillover effects from one province to another without using regions as boundaries to illustrate the reason and path for the spatial spillover effects of educational factors.

## Supporting information

**S1 File. Geographic distance spatial weight matrix.**
(XLSX)

**S2 File. Dataset.**
(XLSX)

## Author Contributions

**Conceptualization:** Xiuhong Li.

**Data curation:** Congying Ma.

**Formal analysis:** Congying Ma.

**Investigation:** Hongchao Wu.

**Methodology:** Congying Ma, Hongchao Wu.

**Resources:** Congying Ma.

**Software:** Hongchao Wu.

**Writing – original draft:** Congying Ma, Hongchao Wu, Xiuhong Li.

**Writing – review & editing:** Congying Ma.

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
