## [Editor Report · Decision Letter 0]

14 Sep 2023

PONE-D-23-27568Spatial spillover of local general higher education expenditures on sustainable regional economic growth: A spatial econometric analysisPLOS ONE

Dear Dr. Ma,

Thank you for submitting your manuscript to PLOS ONE. After careful consideration, we feel that it has merit but does not fully meet PLOS ONE’s publication criteria as it currently stands. Therefore, we invite you to submit a revised version of the manuscript that addresses the points raised during the review process.

I have four comments:

1. Citations: Throughout the paper, we have noticed that there is a lack of appropriate citations to support your claims and arguments. We kindly request that you revisit your manuscript and make sure to cite relevant literature, including recent publications, to strengthen your arguments.

2. Overall writing: While your paper presents interesting findings, the overall writing could be improved. We recommend that you revise the manuscript to ensure that it is clear, concise, and well-organized. Pay particular attention to the following areas:

  a. Language and grammar: Proofread your manuscript for grammatical errors and awkward phrasing. This will help improve the readability of your paper.

  b. Structure and organization: Ensure that each section of your paper flows logically from one to the next. Make sure that you have clearly outlined your research questions, methods, results, and conclusions.

  c. Clarity of arguments: Ensure that your arguments are presented in a clear and logical manner, and that each point is supported by appropriate evidence from the literature.

3. Discussion: The discussion section needs to be more focused on the implications of your findings. Consider discussing the broader context of your results and how they contribute to the existing body of knowledge. Be sure to address any limitations of your study and suggest avenues for future research.

4. Figures and tables: Ensure that all figures and tables are clear, properly labeled, and referenced within the text. Additionally, provide more detailed captions to help readers understand the information presented.

We look forward to receiving your revised manuscript.

Kind regards,

Difang Huang

Academic Editor

PLOS ONE
---

## [Author Response · Author response to Decision Letter 0]

21 Sep 2023

Dear Reviewer,

Thank you for your comments concerning our manuscript “Spatial spillover of local general higher education expenditures on sustainable regional economic growth: A spatial econometric analysis”. First, we are grateful for the reviewer’s thoughtful comments and suggestions. Your valuable comments have helped us significantly improve our manuscript. We have thoroughly revised our manuscript according to the comments. The revised portions are marked in red in the article, and our responses and the reviewer’s original comments are given below in blue.

Reviewer

Comments to the Author(s)

After careful consideration, we feel that it has merit but does not fully meet PLOS ONE’s publication criteria as it currently stands.

Comment 1: Citations: Throughout the paper, we have noticed that there is a lack of appropriate citations to support your claims and arguments. We kindly request that you revisit your manuscript and make sure to cite relevant literature, including recent publications, to strengthen your arguments.

Response 1: Thank you for your comments. We revisited the manuscript and cited relevant citations to support the arguments, including recent publications. For instance: Line 60, page 3; Line 68, page 4; Line 74, page 4; Line 91, page 5; Line 186, page 9; Line 215, page 11; Line 250, page 13; Line 290, 291, page 15; Line 302, page 16; Line 407, 410, page 23; Line 428, page 24; Line 485, page 27.

Comment 2: Language and grammar: Proofread your manuscript for grammatical errors and awkward phrasing. This will improve the readability of your paper.

Response 2: Thank you for your comments. We carefully checked the language and grammar of the manuscript, and it was deeply polished by AJE. Proof of polish has been submitted as an attachment.

Comment 3: Structure and organization: Ensure that each section of your paper flows logically from one to the next. Make sure that you have clearly outlined your research questions, methods, results, and conclusions.

Response 3: Thank you for your comments. Inspired by the reviewers' comments, we further adjusted the structure and organization. As mentioned in section 1 of the revised manuscript (Lines 112-116, page 6):

The paper is structured as follows. The following section reviews the related and latest research. Section 3 introduces the selection of variables and methods of data analysis. Section 4 reports the spatial characteristics of LGHEs and analyzes the effect of LGHEs on regional economic growth. Section 5 offers conclusions and further discussion. Section 6 discusses the limitations of the study and future work.

In addition, we add transitional language to ensure that each section of the manuscript flows logically from one to the next. For instance, Line 54, page3; Line 61, page3; Line 99, page 5; Line 220-221, page11; Line 429-430, page24; Line 482, page26.

Finally, we also further outline the outlined research questions, methods, results, and conclusions of our study. As mentioned in section 1 of the revised manuscript (Lines 71-111, pages 4-6):

The top priority of this study is to present the spatial characteristics of LGHEs. In describing the key features of regional differences, many scholars have used the Gini coefficient and the Theil index to quantitatively measure regional education expenditure disparities [8,9]. However, conventional statistical techniques have failed to distinguish the spatial differences. In this study, we investigated the spatial characteristics of LGHEs in China based on the exploratory spatial data analysis method (ESDA).

Another important purpose of this study is to explore the effect of LGHEs on regional economic growth. With the development of new economic geography, the role of education expenditure agglomeration in economic development is increasingly important [10]. However, most empirical studies that have investigated higher education expenditure (at its aggregate and disaggregate levels) and the economic growth nexus are based on the independence assumption and ignore the significant spatial interaction effects between the variables [11,12]. This may result in estimate bias [13], which may have led to an underestimation of the impacts of higher education expenditure on economic growth in the existing research. Therefore, we further apply the spatial Durbin model (SDM) to test the impact of LGHEs on regional economic growth.

To account for the spatial spillover effects, Anselin’s spatial theory was used as the theoretical basis for the spatial autocorrelation test, spatial econometric model selection and estimation [13]. Empirically, we investigated the spatial characteristics of LGHEs in China and tested the theoretical hypothesis that LGHEs can be influenced by neighboring provinces. Then, we used the SDM to calculate the direct and spillover effects of education expenditures on growth in different regions. Furthermore, the short- and long-term effects by which LGHEs promote economic growth were explored using the DSDM. An exploration of these issues is of great value for promoting educational equity and driving sustainable regional economic growth.

Different from previous studies, we offer two important contributions to the literature. First, we used ESDA to investigate the global and local spatial correlation characteristics of LGHEs. The ESDA method compensates for the defect of classical statistics that ignores geospatial elements and presents differences (spatial agglomeration or heterogeneity) in spatial distribution from a geospatial perspective [14,15]. Notably, LGHEs in a spatial agglomeration state are a premise for studying the spatial spillover of expenditures. Second, we built a more accurate SDM and dynamic spatial Durbin model (DSDM) with spatiotemporal fixed effects, which embrace spatially lagged terms, both dependent and independent variables, drawing on the complementary strengths of the spatial error model (SEM) and spatial lag panel (SLM) [13]. Since the literature concerning regional economic growth is rare, we used the models to investigate the direct and indirect effects and short- and long-term effects of LGHEs on the economic growth of different regions in China.

Comment 4: Clarity of arguments: Ensure that your arguments are presented in a clear and logical manner and that each point is supported by appropriate evidence from the literature.

Response 4: Thank you for your comments. We present the arguments in a clear and logical manner and cite relevant citations to support the arguments, including recent publications. For instance: Line 60, page 3; Line 68, page 4; Line 74, page 4; Line 91, page 5; Line 186, page 9; Line 215, page 11; Line 250, page 13; Line 290-291, page 15; Line 302, page 16; Line 407, 410, page 23; Line 428, page 24; Line 485, page 27.

Comment 5: Discussion: The discussion section needs to be more focused on the implications of your findings. Consider discussing the broader context of your results and how they contribute to the existing body of knowledge. Be sure to address any limitations of your study and suggest avenues for future research.

Response 5: Thank you for your comments. We revised the discussion section, further highlighted the contribution and significance of our study, and proposed relevant policy implications. In addition, we also address the limitations of the study and suggest avenues for future research. The suggestions are applicable not only for China but also for other countries. As mentioned in sections 5 and 6 of the revised manuscript (lines 467-496, pages 26-27):

Section 5: The contributions of this paper are as follows. First, we use EDSA to analyze the spatial characteristics of LGHE and verify the theoretical hypothesis that LGHEs have spatial autocorrelation. Second, spatial econometric analysis compensates for the ignorance of the spatial spillover effects of variables when exploring the relationship between the LGHE and regional economic growth. Finally, this paper expanded the application of spatial theory in the field of economic growth.

The findings are of great value to promoting fair investments in local general higher education, reducing regional disparities, and driving sustainable regional economic growth. They may also have significant implications worldwide for optimizing the allocation of scarce resources for education expenditures and the promotion of sustainable regional economic growth. In particular, our findings may provide useful suggestions for countries with regional economic growth disparities or that are experiencing the transformation of economic development patterns (e.g., from traditional to sustainable growth).

Section 6: There are still some limitations in this study. We focus on LGHEs as the independent variable and do not consider the influence of other educational factors (e.g., per student schooling years) on economic growth. In fact, human capital and physical capital jointly affect economic growth [18]. Therefore, educational factors should be considered comprehensively to improve the scientific rationality of measuring the impact of local general higher education on economic growth. This study also lacks comparative studies across multiple countries.

In the future, there is still much pertinent work to do. For instance, the influence of changes in higher education expenditure inequality on economic growth should be discussed across multiple countries. Future work may also explore how spillover effects of other educational factors (e.g., per student schooling years) affect the sustainable growth of regional economies. Researchers can further expand the analysis of regional spillover effects from one province to another without using regions as boundaries to illustrate the reason and path for the spatial spillover effects of educational factors.

Comment 6: Figures and tables: Ensure that all figures and tables are clear, properly labeled, and referenced within the text. Additionally, provide more detailed captions to help readers understand the information presented.

Response 6: Thank you for your comments. We have improved the format of all figures using Preflight Analysis and Conversion Engine (PACE) digital diagnostic tool and tables to ensure that they are clear, properly labeled, and referenced within the text. In addition, we provide more detailed captions. For instance: Line 262, page 14; Line 284, page 15; Line 306, page 16; Line 312, page17; Line 342, page 19; Line 365, page 20; Line 389, page 22. Finally, we drew the LISA cluster diagram of LGHEs (Fig 1), and further revise the figure using Preflight Analysis and Conversion Engine (PACE) digital diagnostic tool. As mentioned in section 4 of the revised manuscript (Line 283, page 15): 

Fig 1. LISA clustering results of LGHEs. a. 2000, b. 2007, c. 2014, d. 2021.

References:

8.Akita, T. Educational expansion and the role of education in expenditure inequality in indonesia since the 1997 financial crisis. Social Indicators Research. 2017; 130(3):1165–1186. https://doi.org/10.1007/s11205-015-1208-y

9.Varughese AR, Bairagya I. Interstate variation in household spending on education in India: Does it influence educational status?. Structural Change and Economic Dynamics. 2021; 59:405–415. https://doi.org/10.1016/j.strueco.2021.09.015

10.LeSage JP, Pace RK. Introduction of spatial econometrics. 1st ed. Boca Raton: CRC Press; 2009.

11.Ojha VP, Ghosh J, Pradhan BK. The role of public expenditure on secondary and higher education for achieving inclusive growth in India. Metroeconomica. 2022; 73(1):49–77. https://doi.org/10.1111/meca.12353

12.Zhu TT, Peng HR, Zhang YJ. The influence of higher education development on economic growth: Evidence from central China. Higher Education Policy. 2018; 31(2):139–157. https://doi.org/10.1057/s41307–017–0047–7

13.Anselin L. Thirty years of spatial econometrics. Papers in Regional Science. 2010; 89(1):3–25. https://doi.org/10.1111/j.1435–5957.2010.00279.x

14.Wang F, Chai W, Shi XT, Dong MR, Yan B. Does regional financial resource contribute to economic growth? From the perspective of spatial correlation network. SAGE Open. 2021; 11(1). https://doi.org/10.1177/2158244021999381

15.Ma SJ, Li L, Ke HM, Zheng YL. Environmental protection, industrial structure and urbanization: Spatiotemporal evidence from Beijing–Tianjin–Hebei, China. Sustainability. 2022; 14(2). https://doi.org/10.3390/su14020795

18.Ding S, Knight J. Why has China grown so fast? The role of physical and human capital formation. Oxford Bulletin of Economics and Statistics. 2011; 73(2):141–174. https://doi.org/10.1111/j.1468–0084.2010.00625.x

We earnestly appreciate your warm work, and we offer special thanks to you for your valuable comments.

We tried our best to improve the original manuscript and made significant changes in the revised manuscript. These changes will not influence the content of the article. Here, we earnestly appreciate the reviewer’s warm work and hope that the connection will meet with approval. In the meantime, we appreciate the editor’s invitation to modify our manuscript. Thank you for giving us the chance to submit the revised manuscript.

Once again, thank you very much for your comments and suggestions.

Best wishes,

Authors of this article

---

## [Decision Letter · Decision Letter 1]

24 Sep 2023

PONE-D-23-27568R1Spatial spillover of local general higher education expenditures on sustainable regional economic growth: A spatial econometric analysisPLOS ONE

Dear Dr. Ma,

Thank you for submitting your manuscript to PLOS ONE. After careful consideration, we feel that it has merit but does not fully meet PLOS ONE’s publication criteria as it currently stands. Therefore, we invite you to submit a revised version of the manuscript that addresses the points raised during the review process.

We look forward to receiving your revised manuscript.

Kind regards,

Difang Huang

Academic Editor

PLOS ONE

Journal Requirements:

Reviewers' comments:

Reviewer's Responses to Questions

**Comments to the Author**

1. If the authors have adequately addressed your comments raised in a previous round of review and you feel that this manuscript is now acceptable for publication, you may indicate that here to bypass the “Comments to the Author” section, enter your conflict of interest statement in the “Confidential to Editor” section, and submit your "Accept" recommendation.

Reviewer #1: All comments have been addressed

Reviewer #2: All comments have been addressed

2. Is the manuscript technically sound, and do the data support the conclusions?

Reviewer #1: Yes

Reviewer #2: Partly

3. Has the statistical analysis been performed appropriately and rigorously? 

Reviewer #1: N/A

Reviewer #2: N/A

4. Have the authors made all data underlying the findings in their manuscript fully available?

Reviewer #1: No

Reviewer #2: No

5. Is the manuscript presented in an intelligible fashion and written in standard English?

Reviewer #1: Yes

Reviewer #2: No

6. Review Comments to the Author

Reviewer #1: Firstly, I would like to suggest improving the literature review section by incorporating relevant papers. Specifically, the following papers may provide valuable insights and enhance the overall quality of your manuscript:

1. Bao, Z., & Huang, D. (2020). "Gender differences in reaction to enforcement mechanisms: A large-scale natural field experiment." This paper explores gender differences in response to enforcement mechanisms, which could be relevant to understanding the impact of education expenditures on regional economic growth. By considering potential gender-specific effects, you can provide a more comprehensive analysis of the relationship between education expenditures and economic outcomes.

2. Chen, M., Huang, D., & Wu, B. (2022). "Interlocking Directorates and Firm Performance: Evidence from China." This study investigates the relationship between interlocking directorates and firm performance, which could be relevant to understanding the mechanisms through which education expenditures influence regional economic growth. By considering the role of corporate governance and inter-firm connections, you can provide a more nuanced analysis of the impact of education expenditures.

Incorporating these papers into your literature review will help situate your study within the existing literature and provide a more comprehensive understanding of the topic.

In addition to improving the literature review, I have a few suggestions to enhance the overall quality of your manuscript:

1. Clarify the research question and objectives: It would be helpful to explicitly state the research question and objectives of your study. This will provide a clear roadmap for the readers and help them understand the purpose and significance of your research.

2. Provide more details on the methodology: While you have mentioned the use of spatial theory and the spatial Durbin model, it would be beneficial to provide more details on the specific methods and techniques employed. This will enable readers to better understand the analytical approach and replicate your study if desired.

3. Discuss the implications of your findings: In the discussion section, it would be valuable to discuss the practical implications of your findings. How can policymakers use the insights from your study to inform their decision-making? What are the potential implications for educational policies and regional development strategies? Providing a clear discussion of the implications will enhance the relevance and impact of your research.

Overall, your manuscript has the potential to make a valuable contribution to the field. By addressing the above-mentioned points, you can significantly improve the clarity, rigor, and impact of your study.

Please find the reference section for the suggested papers below:

References:

Bao, Z., & Huang, D. (2020). Gender differences in reaction to enforcement mechanisms: A large-scale natural field experiment.

Chen, M., Huang, D., & Wu, B. (2022). Interlocking Directorates and Firm Performance: Evidence from China.

Thank you for considering these suggestions. I look forward to receiving the revised version of your manuscript.

Reviewer #2: (No Response)

7. PLOS authors have the option to publish the peer review history of their article (what does this mean?). If published, this will include your full peer review and any attached files.

Reviewer #1: No

Reviewer #2: No

---

## [Author Response · Author response to Decision Letter 1]

27 Sep 2023

Dear Reviewer,

Thank you for your comments concerning our manuscript “Spatial spillover of local general higher education expenditures on sustainable regional economic growth: A spatial econometric analysis”. First, we are grateful for the reviewer’s thoughtful comments and suggestions. Your valuable comments have helped us significantly improve our manuscript. We have thoroughly revised our manuscript according to the comments. The revised portions are marked in red in the article, and our responses and the reviewer’s original comments are given below in blue.

Journal Requirements

Response: Thank you for your comments. We reviewed the reference list and ensured that it was complete and correct. We do not cite papers that have been retracted. Any changes to the reference list have been mentioned in the rebuttal letter that accompanies the revised manuscript. For instance, Lines 540, 544, 546, 550, page 30; Lines 568, 579, page 31; Lines 583, 599, page 32; Lines 603, 606, page 33; Lines 623, 632-637, page 34; Line 683, page 36; Lines 696-697, 703, page 37; Line 706, page 37; Line 718, page 38; Line 732, page 39.

Reviewer #1

Comments to the Author(s)

Overall, your manuscript has the potential to make a valuable contribution to the field. By addressing the above-mentioned points, you can significantly improve the clarity, rigor, and impact of your study.

Comment 1: Firstly, I would like to suggest improving the literature review section by incorporating relevant papers. Specifically, the following papers may provide valuable insights and enhance the overall quality of your manuscript:

1. Bao, Z., & Huang, D. (2020). "Gender differences in reaction to enforcement mechanisms: A large-scale natural field experiment." This paper explores gender differences in response to enforcement mechanisms, which could be relevant to understanding the impact of education expenditures on regional economic growth. By considering potential gender-specific effects, you can provide a more comprehensive analysis of the relationship between education expenditures and economic outcomes.

2. Chen, M., Huang, D., & Wu, B. (2022). "Interlocking Directorates and Firm Performance: Evidence from China." This study investigates the relationship between interlocking directorates and firm performance, which could be relevant to understanding the mechanisms through which education expenditures influence regional economic growth. By considering the role of corporate governance and inter-firm connections, you can provide a more nuanced analysis of the impact of education expenditures.

Incorporating these papers into your literature review will help situate your study within the existing literature and provide a more comprehensive understanding of the topic.

Response 1: Thank you for your comments. Accordingly, we incorporated the abovementioned papers into the literature review section. By considering potential gender-specific effects and the role of corporate governance and interfirm connections, we provide a more comprehensive analysis of the relationship between education expenditures and economic outcomes. As mentioned in Section 2 and reference section of the revised manuscript (Lines 141-144, page 7; Lines 632-637, page 34):

Furthermore, its economic growth effects are affected by gender differences in reactions to enforcement messages, as well as by governance structures and interorganizational connections [32,33]. 

References:

32.Bao ZY, Huang DF. Gender differences in reactions to enforcement mechanisms: A large-scale natural field experiment; 2020 [cited 2023 Sep 26]. Database: SSRN [Internet]. Available from: http://dx.doi.org/10.2139/ssrn.3641282

33.Chen MZ, Huang DF, Wu BY. Interlocking Directorates and Firm Performance: Evidence from China; 2022 [cited 2023 Sep 26]. Database: SSRN [Internet]. Available from: http://dx.doi.org/10.2139/ssrn.4005022

Comment 2: Clarify the research question and objectives: It would be helpful to explicitly state the research question and objectives of your study. This will provide a clear roadmap for the readers and help them understand the purpose and significance of your research.

Response 2: Thank you for your comments. Based on the research background, we propose research questions and further clarify the research objectives of this study. The research questions in this study are as follows: What kind of spatial characteristics do LGHEs present? What impact do LGHEs have on regional economic growth, and how does it affect regional growth? The research objectives of this study are as follows: The top goal of this study is to present the spatial characteristics of LGHEs. Another important purpose of this study is to explore the effect of LGHEs on regional economic growth. As mentioned in Section 1 of the revised manuscript (Lines 69-91, pages 4-5):

Hence, it is necessary to explore the following questions in this study: What kind of spatial characteristics do LGHEs present? What impact does LGHEs have on regional economic growth, and how does it affect regional growth? These issues are important for promoting educational equity and driving sustainable regional economic growth.

The top goal of this study is to present the spatial characteristics of LGHEs. In describing the key features of regional differences, many scholars have used the Gini coefficient and the Theil index to quantitatively measure regional education expenditure disparities [8,9]. However, conventional statistical techniques have failed to distinguish the spatial differences. In this study, we investigated the spatial characteristics of LGHEs in China based on the exploratory spatial data analysis method (ESDA).

Another important purpose of this study is to explore the effect of LGHEs on regional economic growth. With the development of new economic geography, the role of education expenditure agglomeration in economic development is increasingly important [10]. However, most empirical studies that have investigated higher education expenditure (at its aggregate and disaggregate levels) and the economic growth nexus are based on the independence assumption and ignore the significant spatial interaction effects between the variables [11,12]. This may result in estimate bias [13], which may have led to an underestimation of the impacts of higher education expenditure on economic growth in the existing research. Therefore, we further apply the spatial Durbin model (SDM) to test the impact of LGHEs on regional economic growth.

Comment 3: Provide more details on the methodology: While you have mentioned the use of spatial theory and the spatial Durbin model, it would be beneficial to provide more details on the specific methods and techniques employed. This will enable readers to better understand the analytical approach and replicate your study if desired.

Response 3: Thank you for your comments. We provide more details on spatial theory. Anselin’s spatial theory constitutes a comprehensive analysis of spatial econometrics, and we use Anselin’s spatial theory as the theoretical basis for the spatial autocorrelation test, spatial econometric model selection and estimation in the study. As mentioned in Section 1 of the revised manuscript (Lines 92-96, page 5):

As a pioneer of spatial econometrics research, Anselin (2010) put forward his spatial theory, which constitutes a comprehensive analysis of spatial econometrics [13]. To account for the spatial spillover effects, Anselin’s spatial theory was used as the theoretical basis for the spatial autocorrelation test, spatial econometric model selection and estimation.

Furthermore, we provide more details on the use of the spatial Durbin model (SDM) and dynamic spatial Durbin model (DSDM). As mentioned in Section 3 and reference section of the revised manuscript (Lines 253-262, pages 13-14; Lines 267-271, page 14; Lines 696-697, page 37):

SDM: 

If there are spatial lag terms, the regression coefficient of the econometric model cannot directly be used to reflect the marginal effect of independent variables. To judge the spatial spillover effect of LGHEs on regional economic growth more effectively, this study applies the partial differential method to measure the direct and spillover effects in the SDM [10,52], accounting for the spatial correlation information of the dependent variable and the independent variable. Equations 5-7 show the direct, spillover, and total effects of the SDM.

where IN is an identity matrix, k represents k-th explanatory variable, and d andrsum denote two operators that can be used to calculate both the mean diagonal element and the mean row sum of the nondiagonal elements of a matrix [52].

Reference:

52.Elhorst JP. Matlab Software for Spatial Panels. International Regional Science Review. 2014; 37(3):389–405. https://doi.org/10.1177/0160017612452429

DSDM: 

It can also be used to test the existence of endogenous and exogenous interaction effects in the short term along with the long term [10]. Equation (8) presents this model.

Equations 9-14 show the direct, spillover, and total effects of the DSDM in the short-term and long-term scales.

Comment 4: Discuss the implications of your findings: In the discussion section, it would be valuable to discuss the practical implications of your findings. How can policymakers use the insights from your study to inform their decision-making? What are the potential implications for educational policies and regional development strategies? Providing a clear discussion of the implications will enhance the relevance and impact of your research.

Response 4: Thank you for your comments. Inspired by the reviewers' comments, we provide a clear discussion of the implications in the Discussion section. We further clarify the implications of our findings for educational policies and regional development strategies to help policymakers make decisions more effectively. As mentioned in Section 5 of the revised manuscript (Lines 490-495, page 27):

The findings are of great value to informing policymakers to make educational policies and regional development strategies more effectively consider the short- and long-term economic impacts of LGHEs, regional inequalities in LGHEs and the spatial effects of neighboring regions, which can further promote fair investments in local general higher education, reduce regional disparities, and drive sustainable regional economic growth.

References:

8.Akita, T. Educational expansion and the role of education in expenditure inequality in Indonesia since the 1997 financial crisis. Social Indicators Research. 2017; 130(3):1165–1186. https://doi.org/10.1007/s11205-015-1208-y

9.Varughese AR, Bairagya I. Interstate variation in household spending on education in India: Does it influence educational status? Structural Change and Economic Dynamics. 2021; 59:405–415. https://doi.org/10.1016/j.strueco.2021.09.015

10.LeSage JP, Pace RK. Introduction of spatial econometrics. 1st ed. New York: Chapman and Hall; 2009.

11.Ojha VP, Ghosh J, Pradhan BK. The role of public expenditure on secondary and higher education for achieving inclusive growth in India. Metroeconomica. 2022; 73(1):49–77. https://doi.org/10.1111/meca.12353

12.Zhu TT, Peng HR, Zhang YJ. The influence of higher education development on economic growth: Evidence from central China. Higher Education Policy. 2018; 31(2):139–157. https://doi.org/10.1057/s41307-017-0047-7

13.Anselin L. Thirty years of spatial econometrics. Papers in Regional Science. 2010; 89(1):3–25. https://doi.org/10.1111/j.1435-5957.2010.00279.x

32.Bao ZY, Huang DF. Gender differences in reactions to enforcement mechanisms: A large-scale natural field experiment; 2020 [cited 2023 Sep 26]. Database: SSRN [Internet]. Available from: http://dx.doi.org/10.2139/ssrn.3641282

33.Chen MZ, Huang DF, Wu BY. Interlocking Directorates and Firm Performance: Evidence from China; 2022 [cited 2023 Sep 26]. Database: SSRN [Internet]. Available from: http://dx.doi.org/10.2139/ssrn.4005022

52.Elhorst JP. Matlab Software for Spatial Panels. International Regional Science Review. 2014; 37(3):389–405. https://doi.org/10.1177/0160017612452429

We earnestly appreciate your work, and we thank you for your valuable comments.

We tried our best to improve the original manuscript and made significant changes in the revised manuscript. These changes will not influence the content of the article. Here, we earnestly appreciate the reviewer’s work and hope that the revised version will meet your approval. In the meantime, we appreciate the editor’s invitation to modify our manuscript. Thank you for giving us the chance to submit the revised manuscript.

Once again, thank you very much for your comments and suggestions.

Best wishes,

Authors of this article

---

## [Editor Report · Decision Letter 2]

29 Sep 2023

Spatial spillover of local general higher education expenditures on sustainable regional economic growth: A spatial econometric analysis

PONE-D-23-27568R2

Dear Dr. Ma,

We’re pleased to inform you that your manuscript has been judged scientifically suitable for publication and will be formally accepted for publication once it meets all outstanding technical requirements.

Kind regards,

Difang Huang

Academic Editor

PLOS ONE

Additional Editor Comments:

Thank you for submitting your manuscript to our journal. After careful review, we have found your work to be promising. However, we would like to request that you provide additional references to support your claims.

Specifically, we would appreciate it if you could cite some recent studies that have investigated the same or similar research questions as yours. This will help to strengthen the validity and reliability of your findings: 

Bao, Z., & Huang, D. (2021). Shadow banking in a crisis: Evidence from FinTech during COVID-19. Journal of Financial and Quantitative Analysis, 56(7), 2320–2355.

Chen, M., Li, N., Zheng, L., Huang, D., & Wu, B. (2022). Dynamic correlation of market connectivity, risk spillover and abnormal volatility in stock price. Physica A: Statistical Mechanics and Its Applications, 587, 126506.

Chen, M., Wang, Y., Wu, B., & Huang, D. (2021). Dynamic analyses of contagion risk and module evolution on the SSE a-shares market based on minimum information entropy. Entropy, 23(4), 434.

Wu, B., Huang, D., & Chen, M. (2023). Estimating Contagion Mechanism in Global Equity Market with Time-Zone Effect. Financial Management 52, 543–572

We look forward to receiving your revised manuscript with the additional references.

---

## [Editor Report · Acceptance letter]

3 Nov 2023

PONE-D-23-27568R2 

Spatial spillover of local general higher education expenditures on sustainable regional economic growth: A spatial econometric analysis 

Dear Dr. Ma:

I'm pleased to inform you that your manuscript has been deemed suitable for publication in PLOS ONE. Congratulations! Your manuscript is now with our production department. 

Kind regards, 

on behalf of

Prof. Difang Huang 

Academic Editor

PLOS ONE